# Multi-Decision Dynamic Intelligent Routing Protocol for Delay-Tolerant Networks

Yao Xiong and Shengming Jiang *

Colleage of Information Engineering, Shanghai Maritime University, Shanghai 201306, China; xiongyao@stu.shmtu.edu.cn
* Correspondence: smjiang@shmtu.edu

**Abstract:** Delay-tolerant networks face challenges in efficiently utilizing network resources and real-time sensing of node and message statuses due to the dynamic changes in their topology. In this paper, we propose a Multi-Decision Dynamic Intelligent (MDDI) routing protocol based on double Q-learning, node relationships, and message attributes to achieve efficient message transmission. In the proposed protocol, the entire network is considered a reinforcement learning environment, with all mobile nodes treated as intelligent agents. Each node maintains two Q-tables, which store the Q-values corresponding to when a node forwards a message to a neighboring node. These Q-values are also related to the network's average latency and average hop count. Additionally, we introduce node relationships to further optimize route selection. Nodes are categorized into three types of relationships: friends, colleagues, and strangers, based on historical interaction information, and message forwarding counts and remaining time are incorporated into the decision-making process. This protocol comprehensively takes into account the attributes of various resources in the network, enabling the dynamic adjustment of message-forwarding decisions as the network evolves. Simulation results show that the proposed multi-decision dynamic intelligent routing protocol achieves the highest message delivery rate as well as the lowest latency and overhead in all states of the network compared with other related routing protocols for DTNs.

**Keywords:** delay-tolerant network; double Q-learning; node relationships; network resources

## 1. Introduction

Future wireless networks must continually evolve to adapt to changing wireless communication environments. In harsh and complex scenarios such as wilderness exploration and underwater sensor deployments, the high mobility of nodes leads to frequent changes in network topology, resulting in message loss or significant delays. Traditional routing protocols are often unreliable and ineffective in such circumstances. To facilitate networking and communication in these challenging environments, researchers have introduced a specialized network paradigm that relies on the mobility of nodes to relay information: Delay Tolerant Networks (DTNs) [1]. As an innovative communication approach, DTNs provide a new method to address the instability of connections in wireless networks. They no longer depend on the connectivity requirements of traditional networks but instead utilize cognitive data transmission between mobile nodes. In DTNs, traditional network connectivity requirements do not apply, as they rely on opportunistic encounters between mobile nodes for data transmission. This allows nodes to relay messages when they come into contact with each other, even in resource-constrained environments. Store-carry-forward routing [2] is one of the key mechanisms for addressing routing challenges in DTNs. If a node is not connected to other nodes, it stores messages in its buffer until another node enters its communication range. Once this occurs, the message is forwarded to that node, with the hope that intermediate nodes can relay the message to the destination node.

Based on this concept, researchers have proposed various DTN routing protocols. DTN research focuses on the design of message routing and forwarding protocols, which play a

key role in DTN message transmission. There are three main types of routing protocols: historical information-based routing protocols, community-based routing protocols, and reinforcement learning-based routing protocols.

In history-based routing protocols, protocols utilize historical information about nodes to assess the probability of encounters between nodes, enabling the selection of the next-hop node for message forwarding. The Prophet protocol [3] calculates the probability of nodes encountering the destination node based on their historical information and uses this encounter probability as a utility value to evaluate the likelihood of successfully forwarding messages to the destination node. Ref. [4] proposes a privacy-preserving protocol for utility-based routing (PPUR) in Delay Tolerant Networks (DTNs). The protocol aims to address privacy concerns while optimizing routing decisions based on utility. Ref. [5] presents the Flooding and Forwarding History-Based Routing (FFHBR) algorithm, which determines the best relay node by analyzing the node encounter history and deciding whether to flood propagate the message or forward it directly to the target node. Ref. [6] proposed an Enhanced Message Replication Technique (EMRT) for DTN routing protocols, where EMRT dynamically adjusts the number of message replicas to minimize overhead and maximize the delivery rate based on encounter-based routing metrics, network congestion, and capacity.

In terms of community attributes, ref. [7] proposes an opportunistic social network routing algorithm (UADT) based on user-adaptive data transmission that takes into account factors such as user preferences, social relationships, and network conditions to improve the efficiency and effectiveness of data transmission in opportunistic social networks. Ref. [8] introduces an adaptive multiple spray-and-wait routing algorithm based on social circles in delay-tolerant networks (DTNs). The algorithm dynamically adjusts the number of message copies sprayed based on the encounter history and social relationships between nodes. Ref. [9] presents a Hybrid Social-Based Routing (HSBR) protocol that utilizes social characteristics, such as community structure and social similarity, to make forwarding decisions. Ref. [10] presents the Community Trend Message Locking (CTML) routing protocol. The protocol locks messages to specific communities based on their content and trends, improving message delivery efficiency in DTNs. However, these routing protocols struggle to adapt quickly to node mobility and changes in network topology. In recent years, researchers have increasingly recognized the application of reinforcement learning [11] in DTNs. Q-learning [12] is a value iteration-based reinforcement learning algorithm that learns a value function called the Q-function. The double Q-learning proposed in [13] is an enhanced version of the Q-learning algorithm. Existing routing algorithms based on reinforcement learning models often depend on the reward function, which significantly influences the effectiveness of the learning strategy. Depending on different routing optimization criteria, tailored reward functions are designed for use during the training and learning processes. Ref. [14] evaluates the probability of encounter between nodes using the Q-learning algorithm for packet casting in vehicular opportunity networks and decides the choice of next-hop node together with relative velocity. Ref. [15] introduces an adaptive routing protocol based on improved double Q-learning, where the algorithm determines the next-hop node for data packets based on a hybrid Q-value. Although the above-proposed Q-learning-based DTN protocol has good results, the algorithm is difficult to learn sufficiently under the extreme conditions of the network to ensure that the best route can be guaranteed to be found in all situations. Ref. [16] proposes a delay-tolerant network routing algorithm called Double Q-learning Routing (DQLR); it utilizes a double Q-learning algorithm to address the issue of packet delivery delay in DTNs. Ref. [17] proposes a Double Q-learning-based routing protocol for opportunistic networks called Off-Policy Reinforcement-based Adaptive Learning (ORAL). The protocol utilizes a weighted double Q-estimator to make routing decisions, addressing the limitations of existing protocols. Table 1 compares the types and drawbacks of the DTN routing protocols mentioned above.

**Table 1.** Comparison of related routing protocols.

| Name of Protocol | Routing-Based | Shortcomings |
| --- | --- | --- |
| Prophet [3] | history | High drop ratio |
| PPUR [4] | history | Low delivery rate when network resources are limited |
| FFHBR [5] | history | High overhead |
| EMRT [6] | history | Low delivery rate when network resources are limited |
| UADT [7] | community | Buffer congestion |
| SC-AMSW [8] | community | High overhead |
| HSBR [9] | community | High overhead when network resources are limited |
| CTML [10] | community | Performs poorly when the network topology changes rapidly |
| Proposed in [14] | Q-learning | Low delivery rate |
| ARSIDQL [15] | Double Q-learning | Low delivery rate |
| DQLR [16] | Double Q-learning | Low delivery rate when network resources are limited |
| ORAL [17] | Double Q-learning | Low delivery rate when network resources are limited |

On the basis of the above, we employ a reinforcement learning algorithm, double Q-learning, to adapt to the rapid changes in the network, in addition to defining the relationships between nodes in the DTN and synthesizing the information attributes. By performing multi-decision routing for the forwarding nodes based on the above factors, the nodes can more effectively decide whether the next hop will successfully forward the message to the destination node.

Most existing routing protocols lack real-time awareness during network transmission and struggle with efficient message forwarding. In response, we designed a Multi-Decision Dynamic Intelligence (MDDI) routing protocol. This protocol combines reinforcement learning algorithms with node relationships and message attributes to achieve a multi-decision routing selection. Its goal is to maintain good message transmission performance in various states of the network. In MDDI, the Q-values in the Q-tables within the nodes are continuously adjusted with network changes, and a double Q-updating strategy is used to avoid overestimation and provide more accurate Q-values for evaluating the performance of the nodes. Meanwhile, in order to adapt to each state of the network, we also take the node relationship into account to decide the best next hop when routing based on the node interaction information as well as the message attributes to provide the overall performance of the DTNs.

The main contributions of this paper are as follows:

(1) Real-time sensing of network performance. In calculating the reward value and the discount factor, we use the average delay of the network, the average number of hops, and the message attributes of the nodes after forwarding the message as the learning training content.

(2) Introducing node relationships to improve routing decisions. Combining the historical interaction information between nodes and considering the global network, nodes are classified into three types of relationships: friends, colleagues, and strangers.

(3) Reinforcement learning is combined with node relationships as well as message attributes to decide the best next hop during the routing process to achieve dynamic intelligent routing decisions.

The rest of this paper is organized as follows: In Sections 2 and 3, routing decisions in a double Q-learning environment and dynamic decisions based on node relationships in DTNs are described, and a multi-decision dynamic intelligent routing protocol is proposed. The performance of the proposed protocol is evaluated and compared with conventional protocols in Section 4. Section 5 discusses the impact of the proposed protocol and some potential limitations. Finally, Section 6 concludes this work and proposes future research directions. To clearly describe the proposed routing protocols in the following sections, we provide a list of notations and abbreviations in Table 2.

**Table 2.** List of notations and abbreviations.

| Notations/Abbreviations | Description | Notations/Abbreviations | Description |
|---|---|---|---|
| $Q$ | Future rewards | $S$ | State |
| $S'$ | The next state | $A$ | Action |
| $A'$ | The next action | $\alpha$ | Learning rate |
| $Q(S, A)$ | Future rewards of taking the action A to the state S | DTN | Delay Tolerance Network |
| $a, b, c, d, e, j, x, y$ | Node | $T^*$ | The total lifetime of the message |
| $m_d^i$ | Message i, whose destination node is d | $R_a(b, m_d^i)$ | Actual reward for forwarding $m_d^i$ from node a to node b |
| $\gamma_a(m_d^i)$ | Discount factor corresponding to $m_d^i$ in node a | $T_{avg}$ | The average delay in the network at the moment |
| $t_b$ | The time it takes for $m_d^i$ to reach node d after being forwarded to node b | $Hops_b$ | The number of hops it takes for $m_d^i$ to reach node d after being forwarded to node b |
| $Hops$ | Number of hops of the message from the source node to the destination node | $TTL$ | The remaining time to live of the message |
| $\gamma^*$ | Discount factor constant | $CT$ | Number of contacts |
| $ET$ | Duration of encounters | $MS$ | The number of messages successfully forwarded each other |
| $rel$ | Relationship | $FTh$ | Friend threshold |
| $CTh$ | Colleague threshold | $NT$ | Encounter interval |
| MDDI | Multi-Decision Dynamic Intelligent | $N$ | The number of periods during which the nodes have not encountered |
| SDDQ | Single-Decision based on double Q-Learning | DDNR | Dynamic decision based on node relationship |
| $N_a$ | The set of contact nodes for node a | $C(a, b)$ | Contact values for nodes a, b |
| $T$ | Time period of recoding CT,ET MS | $T'$ | Time period of recoding NT |

## 2. Routing Decision in Double Q-Learning Environment

In this section, we will explore the application of Q-learning algorithms for routing decisions in DTNs. Our main focus will be on two key components: the dual Q-learning algorithm and the Q-table update.

### 2.1. Double Q-Learning Algorithm in DTNs

Q-Learning is a value-based algorithm in the field of reinforcement learning, and its core involves constructing a table called the Q-table. In this table, each row corresponds to a state, each column corresponds to an action, and the value in each cell represents the maximum expected future reward for taking a specific action in a particular state. By progressively updating the Q-Table, the algorithm can gradually learn the optimal action strategy for each state. In this way, by selecting the action with the maximum value in the corresponding row for each state, the goal of maximizing cumulative rewards can be achieved. The Q-learning model iteratively trains through various components, including agents, states, actions, rewards, episodes, and Q-values. The update formula for the state-action values in the Q-learning algorithm is shown in (1). $S$ and $A$ represent a state and an action, respectively. Q represents the Q-value for taking action $A$ in state $S$, $S'$ represents the next state, $A'$ represents the next action, $R$ represents the actual reward obtained from taking that action, $\alpha$ is the learning rate, $\gamma$ is the discount factor, and $\max(Q(S', A'))$ represents the Q-value of the action with the highest Q-value among all possible actions in state $S$.

$$Q(S, A) = Q(S, A) + \alpha(R + \gamma \max(Q(S', A')) - Q(S, A)) \tag{1}$$

The significance of (1) is to gradually establish the optimal policy by updating the Q-value in the Q-table for the action that yields the highest reward in the current state $S$. The discount factor $\gamma$ takes into account future rewards, making the algorithm more focused on

long-term benefits rather than short-sighted strategies. It also results in smoother updates of Q-values, avoiding abrupt changes. It can be observed that the Q-learning algorithm's update is a typical application of temporal difference methods.

During the message delivery process from source nodes to destination nodes, the entire network provides the necessary information for message forwarding. Therefore, in this paper, we consider the entire mobile network as a reinforcement learning environment, where nodes that relay messages are treated as intelligent agents. The action space of these agents consists of forwarding data packets to the next-hop nodes, and selecting the appropriate next-hop to forward the message is considered an action choice in reinforcement learning. All nodes in the network can serve as storage nodes for data packets, so the collection of all nodes in the network forms the state space of the agents. When a message is successfully forwarded to the next-hop node, the agent receives an immediate reward value from the environment, which is used for updating the Q-values.

Figure 1 depicts a typical reinforcement learning example within a delay-tolerant network. In this illustration, a message resides at node $a$, which can be regarded as a state. $m_d^i$ represents message $i$ whose destination node is $d$. Nodes $b$, $c$, and $e$ represent the next action choices made by node $a$ to forward $m_d^i$ to the destination node $d$. The Q-table stores the Q-values associated with node $a$ by choosing nodes $a$, $b$, and $e$ as forwarding nodes for $m_d^i$. When the message eventually reaches its destination node $d$, we update the rewards associated with node $a$ of choices of nodes $a$, $b$, and $e$ as forwarding nodes for $m_d^i$.

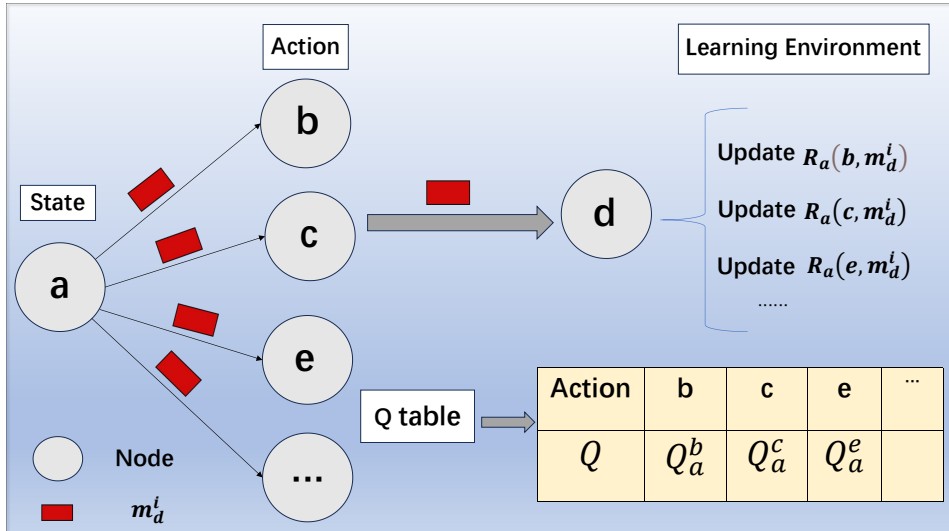

**Figure 1.** Q-learning in DTN.

In the proposed protocol, each node in the network stores and maintains two Q-tables, denoted as $Q^A$ and $Q^B$. These Q-tables consist of columns and rows, storing reward lists for the best actions for each state. When a node in the network begins to move without having forwarded any messages yet, it initializes the corresponding Q-values in its Q-tables to 0 upon encountering other nodes. Whenever a message is successfully relayed to the destination node by a node that has relayed the message, it updates its two Q-tables and shares its local information only with neighboring nodes. The two Q-values stored in the respective Q-tables are used for predicting the optimal action for the next step and predicting the optimal action for the termination state (successfully forwarding the message to the destination node). One Q-value is used for selecting the best action, and the other is used to estimate the maximum Q-value. Double Q-learning is employed to address the problem of overestimation that may lead to routing local optima. The two Q-values change with variations in the network's topology and node states, allowing the proposed algorithm model to adapt to highly dynamic environments.

### 2.2. Q-Value Update

In DTNs, the dynamic movement of nodes allows for opportunistic connections between nodes. Messages are transmitted hop by hop towards the destination node, starting from the source node, following routing protocols, and leveraging opportunistic contacts between nodes. During this process, the Q-value update formula is used to establish and update the corresponding state-action values. In this reinforcement learning environment, the learning task is assigned to each node, and as a result, the learning process involves updating the Q-tables; $Q^A$ and $Q^B$ learn from each other, and they represent future reward values for the same action and state with the same update formula but different values. Their update formulas are shown in (2) and (3).

$$Q_a^A(b, m_d^i) = (1 - \alpha)Q_a^A(b, m_d^i) + \\ \alpha(R_a(b, m_d^i) + \gamma_a(m_d^i)\max(Q_a^B(x, m_d^i))) \tag{2}$$

$$Q_a^B(b, m_d^i) = (1 - \alpha)Q_a^B(b, m_d^i) + \\ \alpha(R_a(b, m_d^i) + \gamma_a(m_d^i)\max(Q_a^A(x, m_d^i))) \tag{3}$$

$Q_a^A$ and $Q_a^B$ represent node $a$ selecting node $b$ as the destination node for $m_d^i$, it represents the expected reward value for the next-hop node, which is the Q-value of node $a$ selecting node $b$ for relaying. $\alpha$ represents the learning rate of a node in the network, which is used to control the extent to which Q-values change with dynamic network variations. $R_a(b, m_d^i)$ represents the actual reward obtained by node $b$ after forwarding $m_d^i$ to node $b$. $\gamma_a(m_d^i)$ is the discount factor associated with node $a$ of forwarding of $m_d^i$. By adjusting $\gamma$, we can control the degree to which the node considers short-term and long-term consequences when selecting the next hop. In extreme cases, when $\gamma = 0$, it only considers the current outcome of the action, whether the next hop is the message's destination node. When $\gamma$ approaches 1, it places more emphasis on previous learning results. $\max(Q_a^A(x, m_d^i))$ and $\max(Q_a^B(y, m_d^i))$ refer to the highest expected reward values when node $a$ forwards to all possible nodes, with $x$ and $y$ representing the corresponding relay nodes at that time. $x, y \in N_a$ and $N_a$ represent the set of contact nodes for node $a$, indicating all nodes that node $a$ encounters during its movement. From the equation, it can be observed that Q-value updates primarily depend on the learning coefficient, the actual reward value, and the discount factor.

When $m_d^i$ reaches the destination node $d$, the $R$ values of all nodes along its forwarding path are updated. The actual reward value $R$ reflects the advantages and disadvantages of one-time forwarding. End-to-end delay represents the time it takes from when a data packet is sent from the source node until it is received by the destination node. The magnitude of delay directly impacts the availability range of DTN and is an important metric to consider in the design of routing protocols. The average hop count is the total sum of hops experienced by copies of all messages in DTNs, divided by the total number of messages generated in the network. The average hop count, along with end-to-end delay, reflects the overall performance of routing protocols, with fewer hops and a lower delay indicating more efficient routing protocols. Therefore, in this paper, when calculating reward values, consideration is given to both hop count and delay to control energy consumption. The update formula for the reward value $R$ is defined as follows in (4).

$$R_a(b, m_d^i) = e^{\left(1 + \frac{T_{avg}}{t_b} + \frac{3}{Hops_b}\right)} \tag{4}$$

$T_{avg}$ represents the average delay in the network at the moment. $t_b$ signifies the time it takes for $m_d^i$ to reach the destination node after being forwarded from node $a$ to node $b$. $\frac{T_{avg}}{t_b}$ represents the contribution of node $a$'s choice of node $b$ as the next hop to reducing the average network delay. The smaller the value of $t_b$, indicating a greater contribution to reducing the average network delay, the larger the reward value. $Hops_b$ signifies the hops

it takes for $m_d^i$ to reach the destination node after being forwarded from node $a$ to node $b$. $\frac{3}{Hops_b}$, which denotes the contribution of node $a$'s selection of node $b$ as the next hop to reducing the average number of hops in the network. The smaller value of $Hops_b$ signifies a greater contribution to reducing network overhead, resulting in a higher reward value. In this context, the definition of 3 represents the average number of hops obtained through multiple simulations and emulations of the prophet routing algorithm.

The discount factor will affect the likelihood of selecting the node previously chosen for forwarding. In this paper, the discount factor is defined to be updated when $m_d^i$ is forwarded or reaches the destination node. It is calculated by (5)

$$\gamma_a(m_d^i) = \gamma^* e^{-\left(\frac{Hops}{3} + \frac{T^*}{TTL}\right)} \tag{5}$$

$\gamma^*$ is a discount factor constant, $T^*$ represents the total lifetime of the message, and $TTL$ indicates the remaining time to live for the message. $Hops$ denotes the number of times $m_d^i$ has been forwarded at this time. The larger the value of $Hops$, the more frequently $m_d^i$ will be forwarded, indicating that the efficiency of previously selected forwarding nodes is moderate. The smaller the value of $TTL$, the shorter the remaining lifetime of $m_d^i$, emphasizing that node $a$ should prioritize the results obtained from selecting the current next hop.

Incorporating the relationship between nodes and messages, as well as real-time performance metrics in the network, such as average message hops and average latency, enables the double Q-learning algorithm to accurately assess the corresponding Q-values when nodes forward messages through learning from the network environment.

## 3. Proposed Routing Protocol

In this section, we introduce a dynamic decision-making routing approach based on node relationships and combine it with the algorithms introduced in Section 2 to propose and elaborate a multi-decision dynamic intelligent routing protocol.

### 3.1. Dynamic Decision Based on Node Relationships

In the design of community-based routing schemes, the primary challenge lies in defining the social relationships between nodes and determining the mechanism for message propagation. Due to intermittent connections between nodes, there is a need for opportunistic message routing. In other words, message exchange occurs only when two nodes are within each other's range, and one node has a lower probability of delivery than the other. Node features can serve as metrics for quantifying node social relationships. Previous research has mainly relied on historical encounter information to derive social relationship attributes. However, depending solely on social relationships can lead to an uneven distribution of loads on nodes with higher degrees of connectivity. Moreover, most have primarily focused on node attributes and have not comprehensively considered message attributes and interaction quality. In this paper, when defining network node relationships, we take into account both the global knowledge of nodes and the comprehensive node interaction information. The network node relationships are defined as the following three types:

**Friendship Relationship:** A friend node, compared with other nodes in the network, interacts with the local node more frequently, and the quality of interaction is higher.

**Colleague Relationship:** A colleague node, compared with other nodes in the network, has occasional contact with the local node within a certain timeframe, but the quality of interaction is generally lower, indicating a less intimate connection.

**Stranger Relationship:** A stranger node, compared with other nodes in the network, has minimal or almost no social interaction with the local node.

Figure 2 illustrates that when nodes $a$ and $d$ established a connection at time t0, they also connected at moments t1, t2, t3, and t4 within the past time interval $T$. When they are in the process of connecting, both nodes update their historical interaction information,

including the average number of encounters within a period $T$, the average encounter duration, and the average number of forwarded messages.

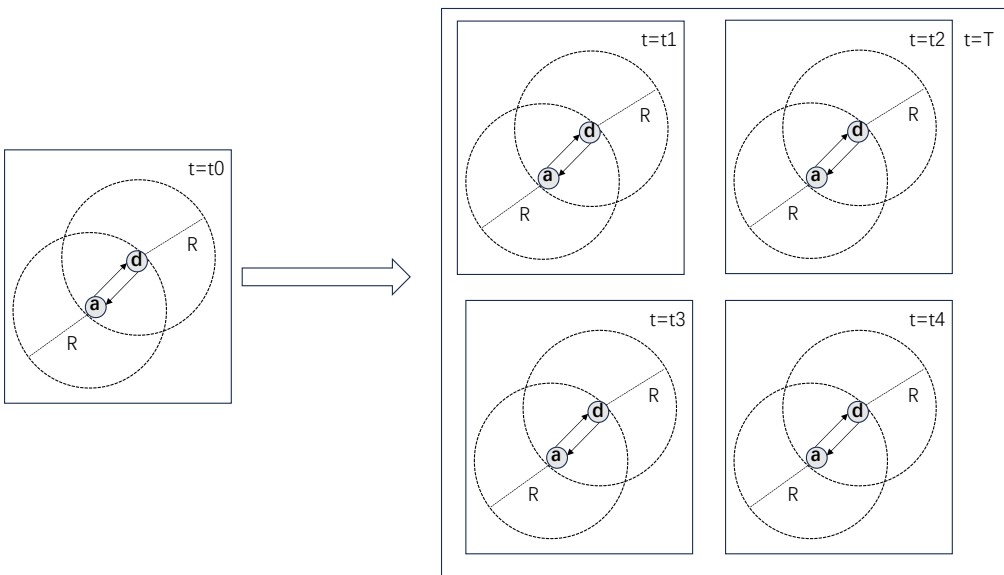

**Figure 2.** The encounter history between node $a$ and node $d$ within the time interval $T$.

Next, let's take nodes $a$, $b$, and $m_d^i$ as an example, suppose we now need to decide whether to forward $m_d^i$ from node $a$ to node $b$ or not. First, the historical encounter information between node $a$ and node $b$ is computed. Node-to-node relationships are determined by historical encounter information and interaction data. Nodes $a$ and $b$ have encounters within the time interval $T$, and each encounter lasts for a duration of $ET_a^b$. The number of encounters is $CT_a^b$. The average encounter duration is as in (6)

$$\overline{ET_a^b} = \frac{\sum_{i=1}^{CT_a^b} ET_a^b}{CT_a^b} \tag{6}$$

The quality of interaction between node $a$ and node $b$ is evaluated based on the number of messages successfully forwarded to each other. The number of messages forwarded at a time is $MS_a^b$. The average message forwarding quantity between nodes $a$ and $b$ within the time interval $T$ is calculated by (7).

$$\overline{MS_a^b} = \frac{\sum_{i=1}^{CT_a^b} MS_a^b}{CT_a^b} \tag{7}$$

Within the time interval $t$, the total sum of historical encounters between node $a$ and other nodes it has come into contact with, the total sum of historical average encounter times, and the total sum of historical average forwarded message quantities are calculated by (8), (9), and (10), respectively.

$$allCT_a = \sum_{j \in N_a} CT_a^j \tag{8}$$

$$all\overline{ET_a} = \sum_{j \in N_a} \overline{ET_a^j} \tag{9}$$

$$all\overline{MS_a} = \sum_{j \in N_a} \overline{MS_a^j} \tag{10}$$

Substitute (8)–(10) into the (11) to calculate the contact value $C(a, b)$.

$$C(a, b) = \frac{CT_a^b}{allCT_a} + \frac{\overline{ET_a^b}}{allET_a} + \frac{\overline{MS_a^b}}{allMS_a} \tag{11}$$

Relationship between node $a$ and node $b$ is defined by (12):

$$rel(a, b) = \begin{cases} friend, & FTh \leq C(a, b) \\ colleague, & CTh \leq C(a, b) < FTh \\ stranger, & 0 \leq C(a, b) < CTh \end{cases} \tag{12}$$

In (12), $FTh$ and $CTh$ represent the friend threshold and colleague threshold, and after multiple simulation verifications, the network performs best when $FTh$ and $CTh$ are set to 0.15 and 0.09, respectively.

Based on node relationships, integrating message remaining time and message forwarding count as one of the decision factors allows for more accurate and efficient forwarding decisions. When nodes $a$ and $b$ establish a connection, if nodes $a$ and $b$ have a friend relationship with the message destination node, then compare their relationship values and forward the message to the node with a greater relationship value.

If nodes $a$ and $b$ have a colleague relationship with the message destination node, not only consider the relationship value but also take into account the message forwarding count, remaining message time, and historical encounter interval. The encounter interval between node $a$ and node $b$ is $NT_a^b$, and their average encounter interval within the period $T'$ ($T' < T$) is calculated by (13), and $n$ is the number of intermittent encounters between them within the period $T'$.

$$\overline{NT_a^b} = \frac{\sum_{i=1}^{n} NT_a^b}{n} \tag{13}$$

If two nodes have not encountered each other for more than one period $T'$, then the historical encounter interval is updated to:

$$\overline{NT_a^b} = (\overline{NT_a^b})_T e^{-N} + T'(1 - e^{-N}) \tag{14}$$

$(\overline{NT_a^b})_T$ represents the average encounter interval within the period $T$. $N$ represents the number of periods during which the nodes have not encountered. $(\overline{NT_a^b})_T$ is equal to 0 when two nodes have not encountered each other for more than one period $T$. Compare the $TTl$ of $m_d^i$ with the historical average encounter interval. If the hop count of $m_d^i$ is less than 3 and the remaining time is greater than the $\overline{NT_a^d}$ and $\overline{NT_b^d}$, then perform a relationship value comparison; otherwise, forward the message to node $b$.

If nodes $a$ and $b$ have a stranger relationship with the message destination node $d$, meaning that the probability of nodes $a$ and $b$ transmitting the message to the destination node is low, then the message is not forwarded and remains at node $a$, awaiting the next connection.

### 3.2. Multi-Decision Dynamic Intelligent Routing Protocol

This section combines the double Q-learning algorithm with node relationships and message attributes to propose a multi-decision dynamic intelligent routing protocol. In this protocol, Q-values and node relationship values serve as the primary decision criteria for nodes to determine whether to forward messages. Additionally, it takes into account changes in message states to ensure effective message transmission. Within the double Q-learning framework, reward computation considers the network's overall average delay and average hops, effectively controlling the number of message duplicates and reducing message delivery latency during the intelligent learning process.

Figure 3 shows the general workflow of the proposed MDDI routing protocol. The MDDI protocol consists of two main parts: a single decision based on double Q-values

(SDDQ) and a dynamic decision based on node relationship (DDNR). SDDQ mainly uses the Q-value in the Q-learning algorithm to make decisions, and the discount factor changes dynamically and is determined by the state of the message in the node. While the actual reward value is updated after the message reaches the destination node. When the connecting node is not the destination node of the message, SDDQ protocol is applied first; if the message cannot be forwarded according to SDDQ protocol, then DDNR is further used. DDNR uses node relationships and message attributes to make a dynamic decision, and the node relationship is derived according to the interaction information between nodes and then integrated with the state of the message at this point in time to make a decision on whether to forward or not. Finally, the Q value and relationship value are updated.

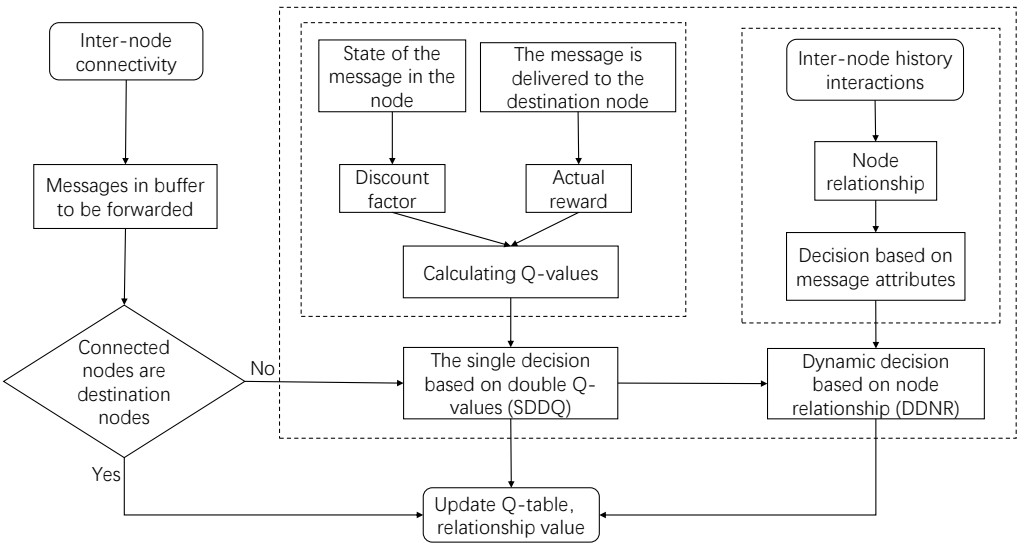

**Figure 3.** Workflow of Proposed MDDI Protocol.

The following section provides a detailed description of the application process of this routing algorithm within the network. In this multi-decision dynamic intelligent routing model, each node in the network stores and updates four tables: two Q-tables, a dynamic node attribute table for participating in routing decisions, and a dynamic message attribute table. The updating of the Q-tables is based on the description in Section 2.2. The dynamic node attribute table contains global information related to the node. It stores encounter information about all the connected nodes for this node, including the number of historical connections within time interval $T$, average connection duration, average message forwarding count, and average encounter interval. The Dynamic Message Attribute Table stores message and node-related attributes, including the time and number of hops that a message is forwarded from that node to the next-hop node until it reaches the destination node, and the discount value used by that node in calculating the Q-value of the node that determines the next-hop node for that message. As an example, let's consider a general node $a$ within the network. As shown in the tables below, In Table 3, $b$, $c$, $d$ denote the nodes that have established a connection with node $a$. The table stores their historical number of encounters with node $a$ during the period $T$, the average encounter duration, the average number of forwarded messages, and the average encounter interval during the period $T'$. In Table 4, $m^e$, $m^f$, $m^g$ are the messages stored in node $a$, and their corresponding dynamic attributes are $\gamma(m)$, $Hop^m$, $t^m$, where $\gamma(m)$ is the discount factor corresponding to different messages, and $Hop^m$ and $t^m$ correspond to the number of hops and time to reach the destination node after the message is forwarded to different nodes

**Table 3.** Dynamic node attribute.

| History | b | c | d |
|---|---|---|---|
| Contact times | $CT_a^b$ | $CT_a^c$ | $CT_a^d$ |
| Connection time | $\overline{ET_a^b}$ | $\overline{ET_a^c}$ | $\overline{ET_a^d}$ |
| Forward count | $\overline{MS_a^b}$ | $\overline{MS_a^c}$ | $\overline{MS_a^d}$ |
| Contact interval | $\overline{NT_a^b}$ | $\overline{NT_a^c}$ | $\overline{NT_a^d}$ |

**Table 4.** Dynamic message attribute.

| Attribute | $m^e$ | $m^f$ | $m^g$ |
|---|---|---|---|
| $Hop^m$ | $Hop_b^{m^e}$ ... $Hop_{j\in N_a}^{m^e}$ | $Hop_b^{m^f}$ ... $Hop_{j\in N_a}^{m^f}$ | $Hop_b^{m^g}$ ... $Hop_{j\in N_a}^{m^g}$ |
| $t^m$ | $t_b^{m^e}$ ... $t_{j\in N_a}^{m^e}$ | $t_b^{m^f}$ ... $t_{j\in N_a}^{m^f}$ | $t_b^{m^g}$ ... $t_{j\in N_a}^{m^g}$ |
| $\gamma(m)$ | $\gamma(m^e)$ | $\gamma(m^f)$ | $\gamma(m^g)$ |

In Figure 4, when nodes a and b establish a connection, they examine their respective dynamic attribute tables (Table 3) when deciding whether to forward a message. They determine their relationship with the message's destination node based on node relationship values. Simultaneously, node a uses the content of its dynamic message attribute table (Table 4) to calculate the corresponding R-value and discount value for forwarding the message to d and compute the corresponding Q-value. The message state is considered at nodes a and b when they share a simultaneous relationship with the destination node.

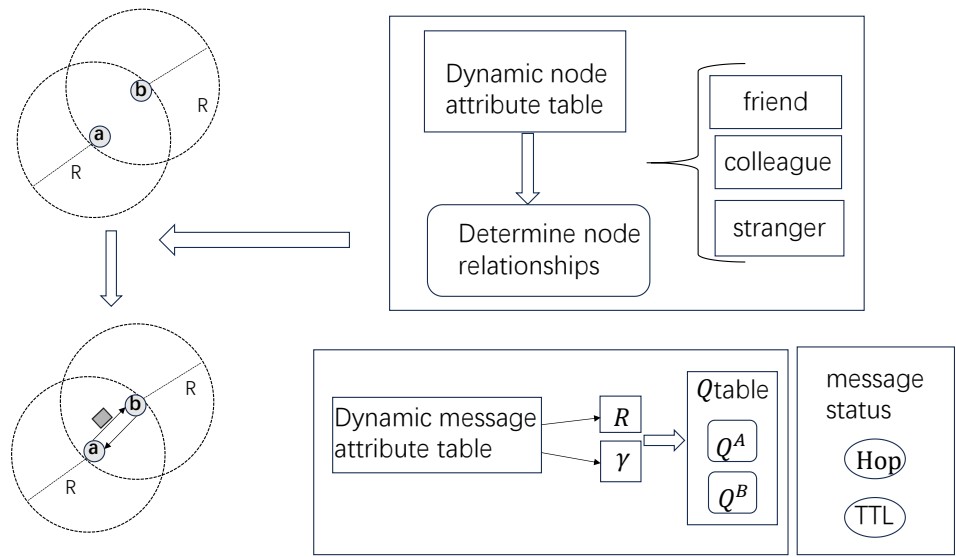

**Figure 4.** Multi-Decision Dynamic Intelligent Routing.

Algorithm 1 provides a detailed description of the specific decision process of this protocol. When node *a* has established a connection with node *b*, the message *m* is pending forwarding within node *a*. First, it is checked whether node *d* is the destination node for message $m_d^i$. If it is not, then a Q-table is randomly selected, and the corresponding Q-value for the connected node is checked to see if it is the maximum. If the Q-table does not exist or the value is not the maximum, a further decision is made based on the node relationships.

---

**Algorithm 1:** Actions when node *a* establishes a connection with node *b*

---

**Input:** Node *a* connects to node *b*

**Output:** Update $Q_{table}$, $C(a,b)$

**1** **for** *The message m destined for node d stored in node a* **do**

**2**  **if** *node b is the destination node for message m* **then**

**3**   | node *a* forwards *m* to node *b*

**4**  **end**

**5**  **else**

**6**   **if** *max $Q_a(x,m) = Q_a(b,m)$* **then**

**7**    | node *a* forwards *m* to node *b*

**8**   **end**

**9**  **end**

**10**  **else**

**11**   **if** *rel(b,d) = friend* **then**

**12**    Compare $C(a,d)$ , $C(b,d)$

**13**    **if** $C(a,d) < C(b,d)$ **then**

**14**     | node *a* forwards *m* to node *b*

**15**    **end**

**16**   **end**

**17**  **end**

**18**  **else**

**19**   **if** *rel(b,d) = colleague* **then**

**20**    **if** *rel(a,d) = colleague* **then**

**21**     **if** $Hops^m < 3$ *and* $avgNT_{T_a}^d < avgNT_{T_b}^d < TTL_m$ **then**

**22**      Compare $C(a,d)$ , $C(b,d)$

**23**      **if** $C(a,d) < C(b,d)$ **then**

**24**       | node *a* forwards *m* to node *b*

**25**      **end**

**26**     **end**

**27**    **end**

**28**    **if** *rel(a,d) = Stranger* **then**

**29**     | node *a* forwards *m* to node *b*

**30**    **end**

**31**   **end**

**32**  **end**

**33** **end**

---

If both nodes have a friend relationship with the destination node, only the relationship values are compared, and the message is forwarded to the node with the higher relationship value. If node *b* has a colleague relationship with the destination node, and node *a* also has a colleague relationship with the destination node, then their contact with the destination node is not frequent enough, so decisions cannot be based solely on encounter information.

The forwarding hop count and remaining message time are compared. If the forwarding count is less than three and the remaining message time is greater than the average encounter interval between the two nodes and the destination node, then further relationship value comparisons are performed. If node *a* and the destination node are strangers, the message is forwarded to node *b*.

If both node *a* and node *b* are strangers to the message destination node, meaning that the probability of both node *a* and node *b* transmitting the message to the destination node is low, then the message is not forwarded and remains in node *a*, awaiting the next connection.

## 4. Simulation Results Evaluation

In this section, we will verify the execution performance of the proposed multi-decision dynamic intelligent routing protocol in the network through experimental simulation and analyze the simulation results in detail.

### 4.1. Simulation Environment

The proposed model in this paper will be simulated and analyzed using the Opportunistic Network Environment Simulator (ONE) [18], which is a simulation tool designed for use in Delay-Tolerant Networking (DTN) environments. The default parameter settings applied in this paper are presented in Table 5 below. The simulation environment includes 126 nodes divided into six groups: two pedestrian groups, one car group, and three tram groups. The pedestrian and car groups consist of 40 nodes each, while the tram groups consist of 2 nodes each. Regarding node movement speeds, the pedestrian groups move at speeds ranging from 0.5–1.5 m/s, the car group moves at speeds ranging from 2.7–13.9 m/s, and the tram groups move at speeds ranging from 7–10 m/s. In terms of motion models, the pedestrian and car groups utilize map-based shortest path motion models, whereas the tram groups use predetermined map-based motion models. In terms of waiting times, the pedestrian groups wait for 0–30 s, the car group waits for 0–60 s, and the tram groups wait for 0–120 s. The communication medium for pedestrian and car groups is a Bluetooth interface with a speed of 250 kb/s and a range of 50 m. The subway train group uses a Bluetooth interface with a transmission range of 800 m and a speed of 20 MB/second. Each experiment is conducted ten times by altering the random seed values for node distribution.

**Table 5.** Simulation parameters setting.

| Parameters | Values |
|---|---|
| Simulation time | 43,200 s |
| Simulation map | Helsinki (4500 km $\times$ 3400 km) |
| Number of nodes | 126 |
| Buffer size | 5 M–50 M |
| Message size | 500 k |
| Moving speed | 0.5 m/s–1.5 m/s |
| Transmission range | 30 m |
| Message generation interval | 25 s–35 s |
| Message TTL | 30 min–90 min |

When evaluating network performance in this paper, several key metrics are employed to assess the quality of the network. These metrics include message delivery rate, network overhead, and average latency [19].

**Message delivery rate:** It measures the success rate of message transmission within the network during the simulation.

**End-to-end average latency**: It calculates the average time it takes for successfully delivered messages to travel from the source node to the destination node during the simulation.

**Network overhead**: It is another crucial metric used to evaluate network performance.

In the double-Q learning algorithm, the learning rate and discount factor constants are crucial parameters for updating Q-values. Consequently, we initially determine the optimal values for these two parameters through experimentation and assess their impact on network performance using the message delivery rate. Figure 5 shows the variation of the delivery rate in MDDI for different $\alpha$ and $\gamma^*$. In Figure 5a, we observe that as the value of $\alpha$ varies within the range of 0 to 1, the message delivery rate initially increases and then decreases. The message delivery rate reaches its maximum when $\alpha$ is set to 0.9. On the other hand, Figure 5b illustrates the behavior of the message delivery rate as the discount factor constant changes within the range of 0 to 1. The message delivery rate exhibits a fluctuating pattern of increase followed by a decrease, with the highest rate occurring at $\gamma^* = 0.8$.

Based on these simulation results, we can conclude that for the MDDI during the simulation, the optimal learning rate is 0.9 and the discount factor constant is 0.8 to achieve the best network performance.

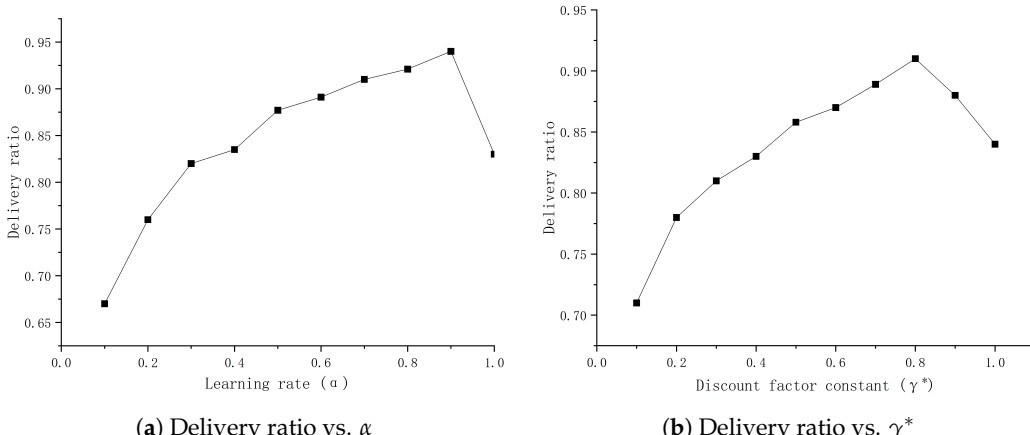

(**a**) Delivery ratio vs. $\alpha$         (**b**) Delivery ratio vs. $\gamma^*$

**Figure 5.** Performance of delivery ratio under different $\alpha$ and $\gamma^*$ in MDDI protocol.

*4.2. Simulation Result*

In this section, we will conduct a network performance evaluation of the proposed MDDI protocol. In addition to comparing it with the typical Epidemic [20] and Prophet [3] routing protocols, Epidemic [20] is based on flooding, with which nodes transmit data packets to every encountered node, resulting in high overhead. We will also compare it with the single decision based on double Q-values (SDDQ) proposed in Section 2 and a dynamic decision based on node relationships (DDNR) proposed in Section 3.1. The values of $\alpha$ and $\gamma^*$ in SDDQ are the same as the values in MMDI. We will assess the impact of variations in network scenarios across three aspects: node caching, message survival duration, and message generation interval, on the network performance of the routing protocols.

**Delivery ratio:** The five protocols in Figures 6a, 7a, and 8a all show an upward trend in delivery rates. The performance of different protocols in terms of delivery rates under various network conditions is described in detail below.

In Figure 6a, MDDI exhibits the most significant advantage, while the Epidemic algorithm performs the worst in terms of delivery rate. The delivery rates of SDDQ and Prophet are comparable, and MDDI outperforms DDNR in terms of delivery performance. SDDQ achieves adaptive learning and can adapt well to the dynamic changes in the network. However, under conditions of limited node cache space, making decisions solely based on Q-values can lead to inaccurate estimations, resulting in average delivery rates in low-cache conditions. DDNR, on the other hand, makes forwarding decisions based on the dynamic attributes of nodes and messages, hence performing well in terms of delivery rates under various cache states. The proposed MDDI not only adapts to dynamic changes in the network through adaptive learning but also dynamically makes decisions based on the attributes of nodes and messages, resulting in the best performance.

In Figure 7a, the delivery rates of the five protocols all exhibit an upward trend and eventually stabilize at a relatively high level. This is because the longer the message's lifetime, the wider the range over which the message is forwarded in the network. Therefore, the majority of messages can be delivered to their destination nodes within their lifetime. DDNR's delivery rate increases as the message's lifetime increases because it takes into account the impact of the message's remaining lifetime when selecting forwarding nodes. Consequently, it achieves a high delivery rate even in scenarios with low message lifetimes. As the message's lifetime increases, both MDDI and SDDQ perform well across various message lifetime states. This is because they quickly find suitable next hops for messages through reinforcement learning. MDDI, which also incorporates node relationships into its decision-making, can find more efficient relay nodes compared with SDDQ, resulting in superior delivery performance.

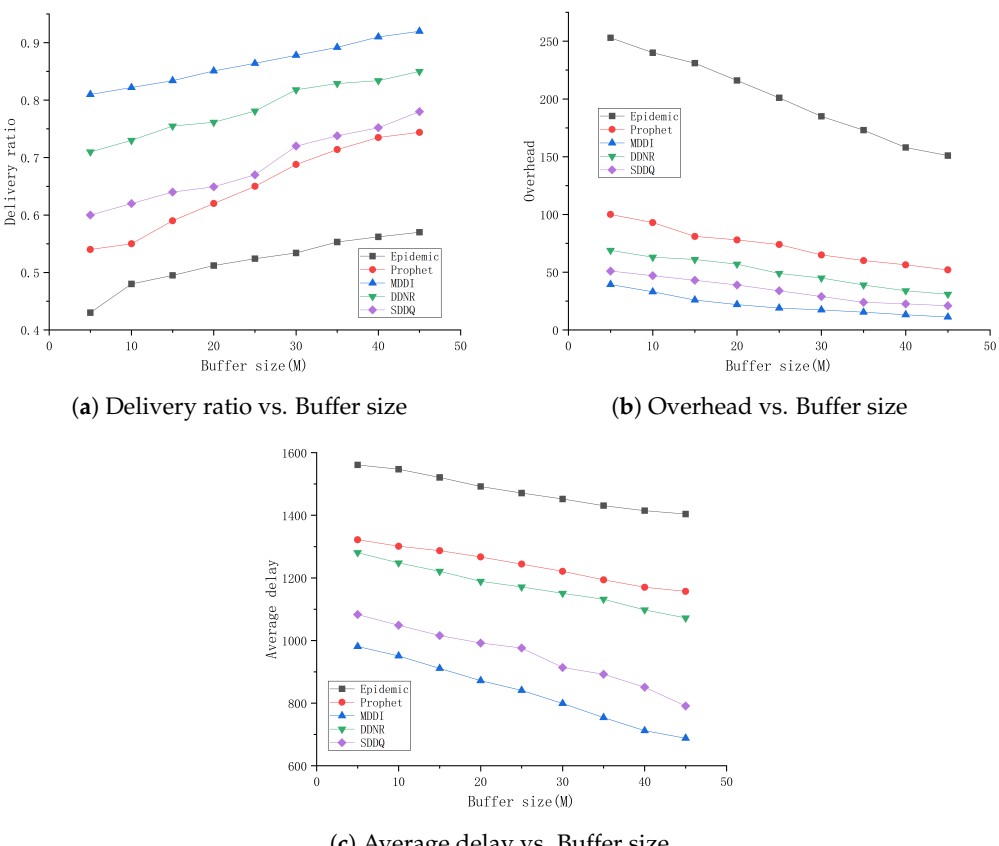

(**a**) Delivery ratio vs. Buffer size

(**b**) Overhead vs. Buffer size

(**c**) Average delay vs. Buffer size

**Figure 6.** Performance of protocols with different Buffer size.

In Figure 8a, at lower message generation intervals, there is a large volume of messages in the network, resulting in high buffer occupancy and an increased risk of congestion, leading to a higher rate of message loss. Consequently, the delivery rates of all five protocols are relatively low during this period. As the message interval increases, the number of messages in the network decreases, and the pressure on the buffers decreases as well. Therefore, in such scenarios, the delivery performance of the Epidemic protocol, which does not limit the number of copies, surpasses that of Prophet. SDDQ performs less effectively in congested network conditions due to its poor learning environment, resulting in a lower delivery rate. However, both MDDI and DDNR maintain high delivery rates even when node buffers are limited because they consider the quality of interactions between nodes, making their node selections more appropriate. Additionally, in scenarios with a lower number of messages in the network, MDDI outperforms DDNR. This is because, under favorable network conditions with suitable reinforcement learning environments, MDDI achieves a higher delivery rate during such periods.

**Overhead:** The network overhead of the five protocols in Figures 7b and 8b all exhibit an increasing trend, while in Figure 6b, they all show a decreasing trend. The performance of different protocols in terms of overhead under various network conditions is described in detail below.

In Figure 6b, DDNR performs less effectively than MDDI and SDDQ because it fails to maintain decision effectiveness when the network topology changes. In contrast, both MDDI and SDDQ use Q-values to decide whether to forward messages. Therefore, with sufficient cache resources, Q-values are updated dynamically with network changes, resulting in lower message loss and effectively reducing network overhead.

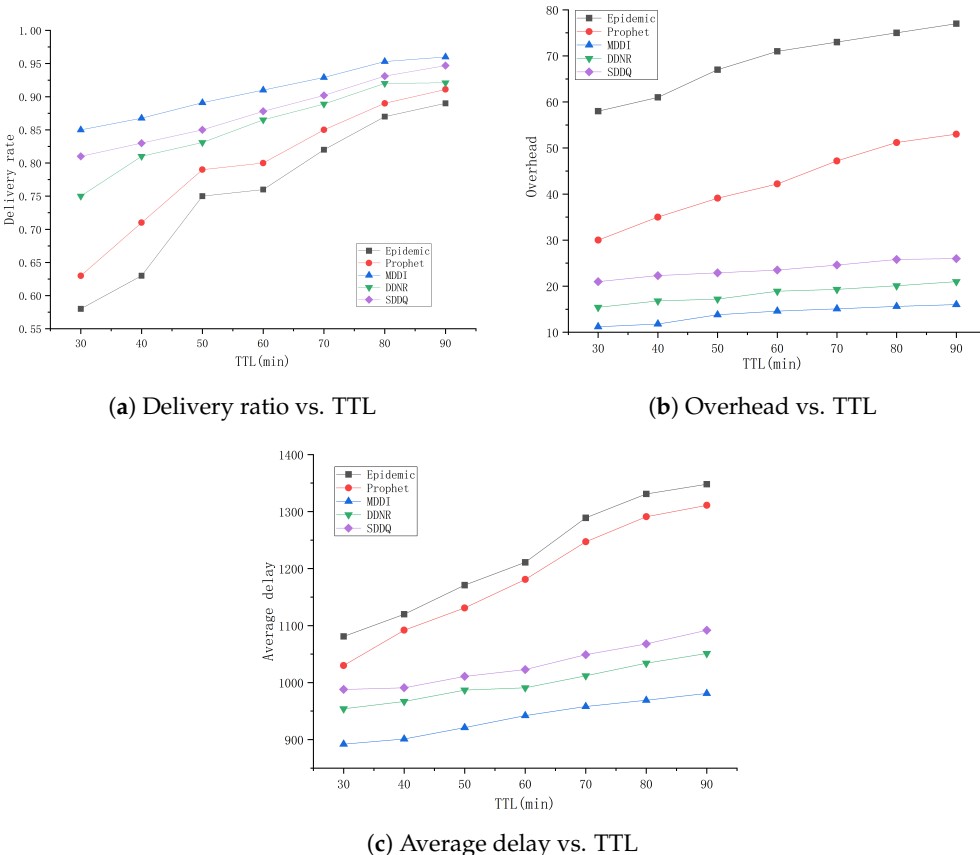

(**a**) Delivery ratio vs. TTL        (**b**) Overhead vs. TTL

(**c**) Average delay vs. TTL

**Figure 7.** Performance of protocols with different TTL.

In Figure 7b, epidemic and prophet exhibit a noticeable upward trend, with significantly higher network overhead compared with the other three algorithms. As message survival time increases, the number of network copies also grows. MDDI, DDNR, and SDDQ all employ strict decision criteria for routing and forwarding, which not only control the number of copies but also increase the probability of messages reaching their destination. Both MDDI and DDNR consider the remaining message survival time when deciding the next hop, which outperforms SDDQ, which relies solely on Q-values. Additionally, MDDI can adapt to network dynamics and learn from them when there is ample message survival time, resulting in lower network overhead compared with DDNR.

In Figure 8b, as the message survival interval increases and the total number of messages decreases, cache resources become more abundant. Consequently, the number of message copies on the network increases. Epidemic shows the most significant increase in network overhead, while Prophet, relying solely on encounter information, also forwards more message copies. SDDQ, when message survival intervals are short and network congestion is high, performs less effectively in deciding the next hop due to limited learning opportunities, resulting in higher network overhead compared with DDNR. However, both MDDI and DDNR consider message forwarding hops, ensuring successful message delivery with fewer intermediate relays when there are many messages in the network. When the total number of messages is low and network resources are abundant, MDDI demonstrates a distinct advantage in dynamic network environments.

**Average delay:** In Figures 6c and 8c, the average latency of the five protocols shows a general decreasing trend, while in Figure 7c, it shows an increasing trend. The performance of different protocols in terms of average delay under various network conditions is described in detail below.

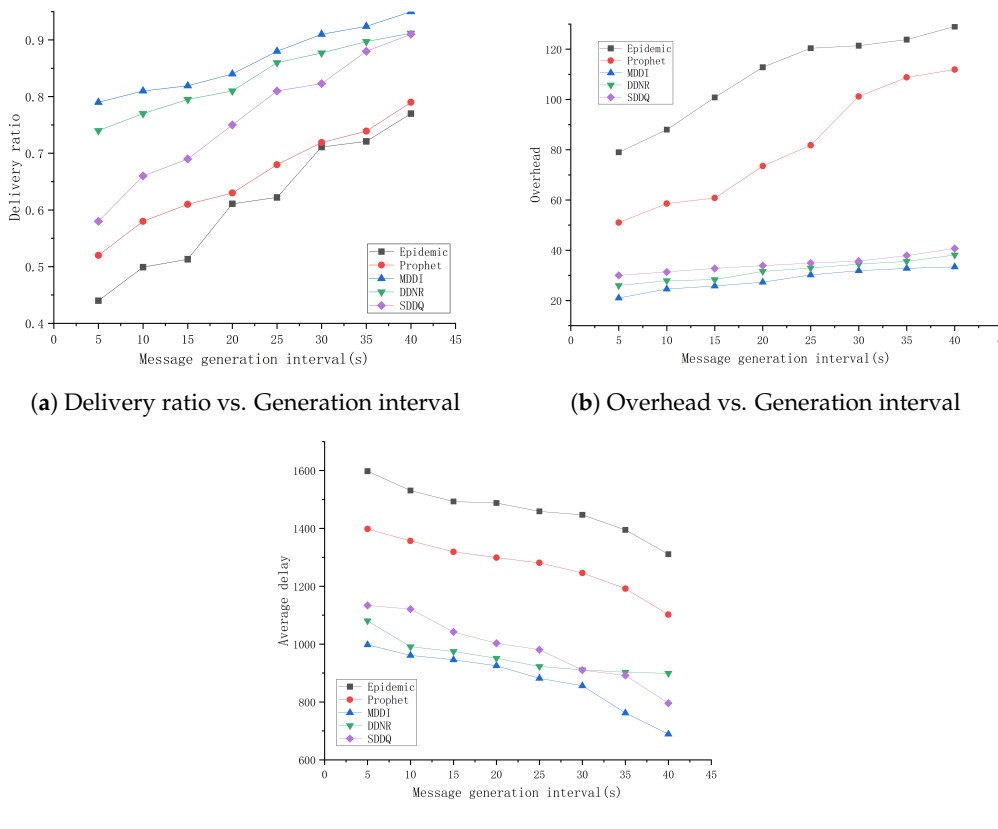

(**a**) Delivery ratio vs. Generation interval  (**b**) Overhead vs. Generation interval

(**c**) Average delay vs. Generation interval

**Figure 8.** Performance of protocols with different Generation interval.

In Figure 6c, although Prophet and DDNR can determine the next forwarding node for messages based on historical information, they cannot find a suitable complete path for messages when the network topology changes. Therefore, their performance in terms of average delay is mediocre. MDDI and SDDQ consider the network's average delay in the discount factor when calculating the Q-value. This not only enables them to effectively find paths to forward messages to their destination nodes by continuously learning in the network but also reduces the overall average delay in the network. Dynamic multi-decision making in MDDI, compared with SDDQ, considers relay node selection more comprehensively and reasonably, resulting in MDDI having the best delay performance.

In Figure 7c, as the message's lifetime increases, messages that are farther from the destination node are successfully delivered, increasing the transmission delay. Epidemic uses a flooding mechanism to forward messages, so when message lifetimes are extended, the number of message copies increases, and they reach the destination node faster, outperforming Prophet, which relies solely on encounter information for decision-making. Both MDDI and SDDQ apply reward values that consider average delay when calculating Q-values, effectively reducing network delay. However, when message lifetimes are shorter, MDDI and DDNR perform better because they both use dynamic attributes of nodes and messages to make decisions, resulting in less time for messages to reach their destination nodes. As message lifetimes increase, MDDI, through effective learning of the network environment, can find more accurate next hops, resulting in the lowest average delay.

In Figure 8c, when message generation intervals are short, there are many messages in the network due to node congestion, resulting in higher average delays. However, when message generation intervals are small, both DDNR and MDDI maintain good performance, as they consider average delay when deciding whether to forward messages. Thus, they can maintain lower average delays even with a high number of messages. As the generation interval increases and the number of messages decreases, with sufficient buffer space,

SDDQ and MDDI can fully learn the network environment, while DDNR excels due to its dynamic decision-making capabilities for selecting relay nodes, resulting in the best delay performance.

## 5. Discussion

In Delay Tolerant Networks (DTN), nodes can only transmit messages through opportunistic connections, posing a challenge in effectively utilizing network resources for message forwarding. In previous protocol studies, achieving both high delivery rates and low overhead has been a difficult balance. Our protocol employs dynamic multi-decision-making, ensuring excellent performance in various network states. In Section 4, validated through simulations, the MDDI routing protocol's ability to fully utilize network resources and enhance message transmission efficiency in mobile network environments is demonstrated.

Compared to other routing protocols introduced in Section 1, our proposed protocol has significant advantages in multiple aspects. Firstly, even in situations where network resources are limited, the MDDI routing protocol maintains high delivery rates and low latency. This protocol utilizes dynamic changes in node and a message attributes during network topology changes, ensuring outstanding performance regardless of whether the network has few nodes or experiences network congestion. Secondly, the MDDI routing protocol consistently incurs lower network overhead. In contrast to some routing protocols discussed in Part Two, which also maintain high delivery rates but generate numerous message duplicates due to flooding, the MDDI routing protocol, through multiple decision-making solutions, precisely selects the next hop for messages, ensuring message delivery to the destination node while effectively controlling the number of message duplicates.

However, despite the numerous advantages of the MDDI routing protocol, there are potential limitations that need consideration. One potential limitation lies in the complex implementation of MDDI routing protocol attributes. Each node in the network needs to maintain four tables (two Q tables, a node attribute table, and message attribute table). In intermittent connection routing environments, these table values require real-time updates, potentially consuming significant node energy. Moreover, frequent interactions between nodes may lead to security issues, making them susceptible to attacks from malicious nodes. For instance, a malicious node might attempt to simulate another node to access its private information.

In addition to the potential limitations discussed above, the protocol has not been widely deployed in practical networks. Implementing this protocol in real-world scenarios may face various challenges. One challenge is that the protocol might not always maintain high performance in different scenarios. For instance, in disaster and emergency rescue scenarios, historical connections among nodes might not follow strong patterns, and urgent messages should be assigned a higher security level, not solely based on encounter history. Another challenge is privacy and security. In military scenarios, selecting nodes with high trust levels for data transmission is crucial.

Addressing these challenges requires a comprehensive consideration of the characteristics of different scenarios and flexible adjustment of the MDDI protocol's parameters and strategies. In disaster and emergency rescue scenarios, adopting more flexible message priority algorithms ensures the timely transmission of urgent messages. In military scenarios, establishing trust models, selecting trustworthy nodes for data transmission, and enhancing network encryption and identity verification measures can improve data security.

To sum up, the introduction of the MDDI routing protocol not only underwent thorough theoretical exploration but also received validation through practical simulations. Its advantages, including high delivery rates, low latency, and minimal network overhead, make it an ideal choice in mobile network environments. These research findings not only supplement the existing knowledge base but also provide valuable insights for the development of more advanced network systems in the future.

## 6. Conclusions

In this paper, we propose a Multi-Decision Dynamic Intelligent (MDDI) routing algorithm for delay-tolerant networks. The algorithm fully considers the characteristics of nodes interacting with information and messages and the variability of network topology. Firstly, we introduce the intelligent double Q-learning algorithm, enabling nodes to learn throughout the entire network. Each node maintains two Q-tables and decides whether to forward a message based on the Q-values in the tables. If the connected node does not have the maximum Q-value in the Q-table, decisions will be made based on the relationship between the nodes, at which point we first determine the relationship between the nodes, i.e., whether they are friends, colleagues, or strangers, and combine the message attributes to make a final decision. Through multiple decisions and intelligent algorithms, we can effectively utilize various resources of the network and sense the states of nodes and messages in real-time, thus improving the network performance.

The simulation results show that our proposed protocol consistently maintains the highest delivery rate as well as the lowest latency and network overhead for different cache sizes, as well as different message survival times and generation intervals. Moreover, analyzing the result graphs, it can be seen that the network overhead of MDDI remains low in all network states, while the message delivery rate can be high even in extreme cases, such as node congestion and short message survival time.

In the next step of our research, we plan to enhance the cache management component in our subsequent research to minimize message loss. Additionally, we will design a node authentication mechanism to enhance data transmission security. Moreover, we intend to implement a confirmation mechanism to promptly discard duplicates of messages that have already been successfully transmitted in the network.

**Author Contributions:** Supervision, S.J.; writing—original draft: Y.X.; writing—review and ending, Y.X. All authors have read and agreed to the published version of the manuscript.

**Funding:** This work was Supported funded by the Innovation Program of Shanghai Municipal Education Commission of China under Grant No. 2021-01-07-00-10-E00121.

**Data Availability Statement:** The program code used in the research can be obtained from the corresponding author upon request.

**Conflicts of Interest:** The author declares no conflict of interst

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
