# Peer review of "Multi-Decision Dynamic Intelligent Routing Protocol for Delay-Tolerant Networks"

_electronics, doi:10.3390/electronics12214528_

Round 1

Reviewer 1 Report

Comments and Suggestions for Authors

Attached.

Author Response

Thank you for your decision and constructive comments on my manuscript. We have carefully considered the suggestion of Reviewer and make some changes. We have tried our best to improve and made some changes in the manuscript.Please see the detailed response in the attached "response.pdf".

Reviewer 2 Report

Comments and Suggestions for Authors

In the manuscript entitled “Multi-Decision Dynamic Intelligent Routing Protocol for Delay-Tolerant Networks” the authors proposed a Multi-Decision Dynamic Intelligent routing protocol based on certain parameters to achieve message transmission efficiency. I congratulate the authors for their. They need to consider the following points.

1.      Add specific results in the abstract.

2.      Majority of the references are old and authors need to replace some of these with new references. 

3. Limited related work is reviewed. This section should be updated by providing details of more recent references. 

4.      There are spelling mistakes. As an example see section 5.1 line 4.

5.      Update x-axis label from “Learning rate” to “Learning rate (α)” in figure 4 a. repeat for figure 4 b.

6.      Update figure 4 title which is generic at the moment. Should mention the algorithm for which these results are generated. 

7.      Conclusion can be improved by discussing individual results from the graphs.  

Comments on the Quality of English Language

English language is OK but there are spelling and other mistakes which should be corrected. 

Author Response

(The authors gave the same response as above.)

Reviewer 3 Report

Comments and Suggestions for Authors

Dear Authors,

Please review the attached document for comments and suggestions. 

Comments on the Quality of English Language

Revise the paper after final edits for spelling, grammar, and flow.

Author Response

Thank you for your decision and constructive comments on my manuscript. We have carefully considered the suggestion of Reviewer and make some changes. We have tried our best to improve and made some changes in the manuscript.Please see the detailed response in the attached "respons.pdf".

Round 2

Reviewer 1 Report

Comments and Suggestions for Authors

The revised version of the manuscript is highly improved over the original one. Unfortunately, there is no separate Authors’ reply, and in the text, I cannot identify the parts, that dispel my doubts raised, particularly in the last point of my previous remarks. 

The new remark, that augments the previous ones:

Eqs (4) & (5) mentioned previously, depend on the mysterious constant “3”, which looks like topology dependent value. Additionally, $T_{avg}$ is unknown for the node. The justification is necessary.

Author Response

Dear Reviewer, I sincerely apologize for the oversight in my previous submission, which led to your not receiving the revised response document. I have rectified this error and resubmitted the response document, which is now attached to this email. In this document, I have clearly marked the modifications made to the manuscript as per your valuable suggestions. Additionally, I have addressed the specific points raised in your latest review.
I appreciate your time and effort in reviewing my manuscript. I kindly request you to kindly review the attached response document, and I am more than willing to provide any further clarification if necessary.
Thank you for your understanding and patience.
Please see the attachment.

Reviewer 3 Report

Comments and Suggestions for Authors

Thank you for your revised review, glad that the comments and my suggestions helped. I appreciate the substantial improvements that have been made to the manuscript. Just one comment, try to add what the section is going to discuss under sections 2, 3, and 4 (do not leave them empty). 

Author Response

Dear Reviewer, Thank you very much for carefully reviewing my manuscript and providing valuable comments and suggestions once again. I greatly appreciate your in-depth assessment of my research. Regarding your suggestion to add a discussion of the contents of each section under sections 2, 3 and 4, we have added a brief description of each section below the headings of those sections.

Round 3

Reviewer 1 Report

Comments and Suggestions for Authors

The justifications provided by the Authors show that common/central knowledge is necessary, hence the way the nodes exchange the data about average time or hop count would increase the scientific contribution of the manuscript.